# EndoBind detects endogenous protein-protein interactions in real time

Anke Bill [1✉], Sheryll Espinola[2], Daniel Guthy[3], Jacob R. Haling[2,4], Mylene Lanter[3], Min Lu[2], Anthony Marelli[2], Angelica Mendiola[2], Loren Miraglia [2], Brandon L. Taylor[2], Leonardo Vargas[2], Anthony P. Orth[2] & Frederick J. King [2✉]

We present two high-throughput compatible methods to detect the interaction of ectopically expressed (RT-Bind) or endogenously tagged (EndoBind) proteins of interest. Both approaches provide temporal evaluation of dimer formation over an extended duration. Using examples of the Nrf2-KEAP1 and the CRAF-KRAS-G12V interaction, we demonstrate that our method allows for the detection of signal for more than 2 days after substrate addition, allowing for continuous monitoring of endogenous protein-protein interactions in real time.

[1] Novartis Institute for Biochemical Research, Oncology, Cambridge, MA, USA. [2] Genomics Institute of the Novartis Research Foundation, Assay Development and High Throughput Screening, San Diego, CA, USA. [3] Novartis Institute for Biochemical Research, Oncology, Basel, Switzerland. [4] Present address: Mirati Therapeutics, Inc., Research, San Diego, CA, USA. ✉email: anke.bill@novartis.com; fred.king@novartis.com

Cell signaling is a highly flexible and transient process that relies on a careful balance between inhibitory and activating signals mediated by protein–protein interactions. A variety of methods for monitoring cellular protein–protein interactions have been described that suffer enormous limitations to capture those highly dynamic signaling events in cells. (i) Ectopic, usually transient overexpression risks non-biologically relevant interactions or artificial modulation of the signaling event itself by disturbing the stoichiometric balance. (ii) Analyzing the protein-protein interactions or signaling state of a cell at a specific time point in end-point assays can lead to misinterpretation or oversight of important underlying processes. (iii) Methods relying on the detection of endogenous protein require either high amounts of protein or special instrumentation, making them incompatible with high-throughput applications.

Here, we present two ultra-sensitive methods—RT-bind and EndoBind—that overcome these limitations and are capable of capturing real-time cellular protein–protein interactions and dynamics of exogenously (RT-bind) or endogenously (EndoBind) expressed proteins in high-throughput.

## Results and discussion

**RT-bind enables the detection of proteins in real-time**. Our methods exploit the split-NanoLuciferase-complementation (NanoBit[TM]) system[1] in which the interaction of the two target proteins, each tagged with one part of modified NanoLuciferase, facilitates the formation of the catalytically active NanoBite enzyme, and hence results in a luminescence signal in the presence of its substrate furimazine. For this, the proteins are reciprocally fused to one of the two optimized portions of modified NanoLuciferase (the smaller part, SmB; or the larger part, LgB) via tailored, flexible linkers. When brought into proximity, the binding of SmB to LgB generates the catalytically active NanoBit enzyme. Despite its only recent publication, the NanoBit[TM] system has been successfully used to study protein:-protein interactions of cytoplasmic and membrane proteins at a given timepoint[2–5]. To confer the capacity to monitor interactions over time we turned to the NanoLuciferase pro-substrate RealTime-Glo, which can be added in media and must be metabolized by viable cells before it can serve as a functional substrate for standard NanoLuciferase. We hypothesized that this pro-substrate, intended for use with holo-NanoLuciferase could serve as a long-lived substrate for reconstituted NanoBit, enabling kinetic measurement of tagged proteins as they interact (Fig. 1a). To test our hypothesis, we generated a HEK293T cell line stably expressing Nrf2 and Keap1 tagged with SmB or LgB of the NanoBit enzyme. We chose this protein pair for our study because cellular Nrf2 levels are known to be tightly regulated via its interaction with Keap1 which, together with Cullin3 (Cul3), continually and dynamically targets Nrf2 for protein degradation[6]. HEK293T cells do not harbor mutations in Nrf2 and Keap1 and show good transduction efficiencies; hence they were a suitable cell line for our study. To avoid artifacts typically associated with high overexpression and concomitant substrate

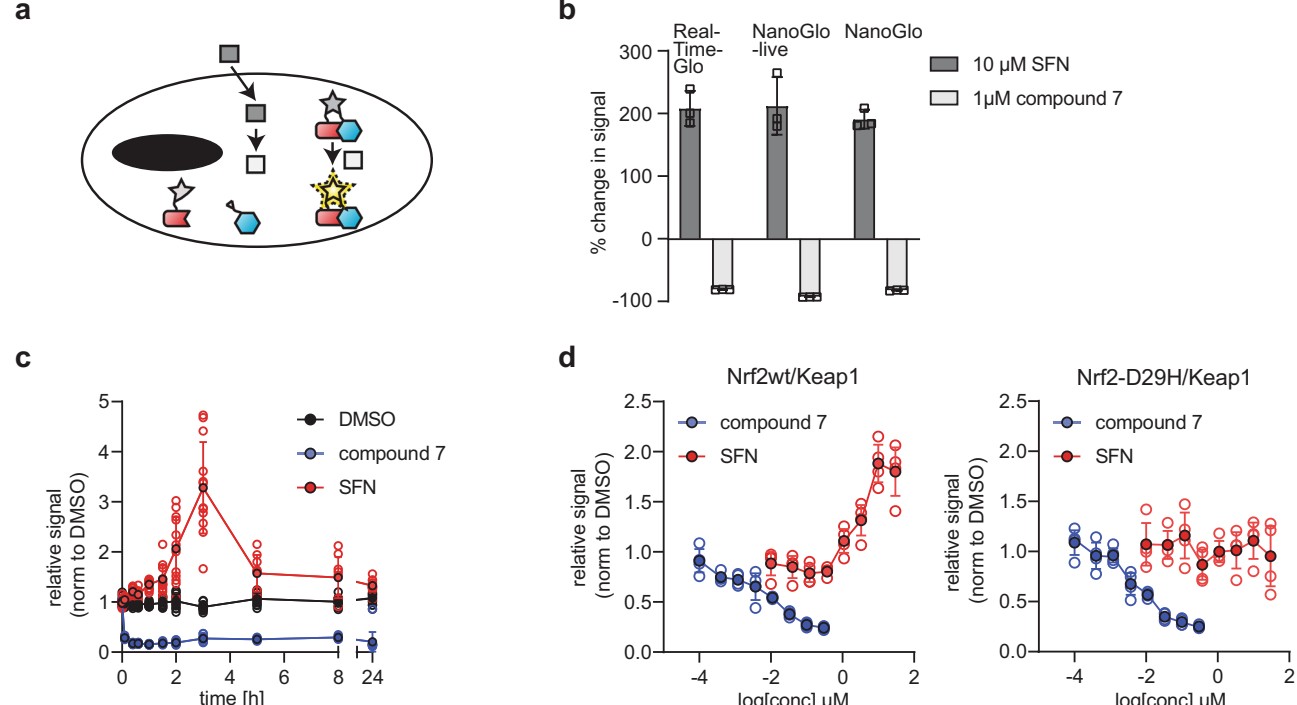

**Fig. 1 Real-time detection of protein interactions in cells on the example of the Nrf2-Keap1 interaction. a** Schematic of RT-Bind principle: cells expressing the protein of interests (red/blue cartoon) tagged with either SmB or LgB of modified NanoLuciferase (gray star), are incubated with a long-lived pro-substrate for NanoLuciferase (dark gray box), which is continuously metabolized by the cells to the NanoLuciferase-compatible substrate (light gray box). Interaction of the proteins of interest leads to complementation of the NanoBit enzyme and, in presence of the metabolized substrate, to luminescence. **b** Relative luminescence signal after 4 h treatment with 10 μM Sulforaphane or 1 μM compound 7 in HEK293T-SmB-Nrf2/Keap1-LgB cells (mean ± s.d., n = 3 wells each, representative experiment) as detected by the use of the indicated luciferase substrates. **c** Relative luminescence signal in HEK293T-SmB-Nrf2/Keap1-LgB cells detected by RT-bind by measuring the same plate at the indicated time points after treatment with 10 μM Sulforaphane or 1 μM compound 7 (mean ± s.d., n = 12 wells each, representative experiment). Data were normalized to the pre-treatment luminescence signal for each well and subsequently normalized to the control treatment (DMSO). **d** Relative luminescence signal in HEK293T-SmB-Nrf2/Keap1-LgB or HEK293T-SmB-Nrf2-D29H/Keap1-LgB cells detected by RT-Bind 4 h after treatment with 10 μM Sulforaphane or 1 μM compound 7 (mean ± s.d., n = 4 wells each, representative experiment). Data were normalized to the control (DMSO) treated sample.

depletion we delivered the constructs for the respective proteins at low MOI via a lentiviral vector employing a shortened EF1-promotor to drive ectopic expression. Stable lines with all eight possible pairwise combinations of reciprocal tag permutations were generated (i.e., N- or C-terminal tagging with SmB or LgB for both Nrf2 and Keap1 and all possible combinations thereof). The cell line with N-terminal SmB-tag on Nrf2 and C-terminal LgB-tag on Keap1 (from herewith referred to as SmB-Nrf2/Keap1-LgB) was chosen for all further experiments based on assay window and expected performance of known Nrf2 modulators (Supplementary Fig. 1a, b). Sulforaphane (SFN), an inhibitor of the Keap1/Cul3-mediated degradation[7], yielded an increase of the Nrf2:Keap1 signal in the SmB-Nrf2/Keap1-LgB cell line and compound 7, an inhibitor of the interaction of Keap1 and Nrf2[8], yielded a decrease. Analysis of Nrf2 levels via western blotting confirmed the overall low level of overexpression of tagged Nrf2 and Keap1, both approximately expressed at twice the levels of the endogenous proteins (Supplementary Fig. 1c). Treatment with SFN led to increased Nrf2 levels in the parental cells as well as in the SmB-Nrf2/Keap1-LgB cell line, demonstrating the dependence of endogenous as well as overexpressed, tagged Nrf2 on degradation by Keap1/Cul3.

As a next step, we tested whether adding the RealTime-Glo substrate to our SmB-Nrf2/Keap1-LgB cell line would allow us to measure the NanoBit signal indicative of the interaction of Nrf2/Keap1. For this, we added the RT-Glo-substrate to the cells at the time of seeding, incubated the cells overnight, added compound on the next morning, and measured the luciferase signal 4 h after compound treatment without the addition of any additional substrate. NanoGlo-live or NanoGlo substrate were added to separate cells as a comparison. The signal measured with the RealTime-Glo substrate demonstrated a significant decrease after treatment with compound 7 and a significant increase after treatment with SFN, similar to the signal changes seen after the addition of the other substrates (Fig. 1b).

In order to investigate whether the use of the RealTime-Glo-substrate would allow us to study the interaction kinetically, we seeded SmB-Nrf2/Keap1-LgB cells in the presence of pro-substrate and incubated the cells overnight. The next day, we measured the basal signal, added either SNF or compound 7, and then re-read the same plate at various time points post compound treatment up to 24 h without the addition of additional reagents or substrate. We observed a near-instant signal decrease in response to the known Nrf2/Keap1 complex disruptor compound 7, an effect sustained over the entire time course of the experiment (Fig. 1c). In contrast, as expected for a compound acting via proteostasis, treatment with SFN increased the luciferase signal steadily until reaching a peak about 4 h after treatment before returning to baseline 8 h after treatment. The slow but steady increase in the luciferase signal is a result of the accumulation of Nrf2 protein due to transient inhibition of degradation by the Keap1/Cul3-complex. To corroborate the specificity of the observed signal, we compared the effect of compound 7 and SFN on our SmB-Nrf2/Keap1-LgB cell line and a similar cell line expressing an Nrf2 mutant insensitive to Keap1-mediated degradation[6] (SmB-Nrf2-D29H/Keap1-LgB, i.e., Nrf2-D29H mutant N-terminally tagged with SmB and Keap1 C-terminally tagged with LgB) (Fig. 1d). The D29H mutant diminishes interaction at one of the two critical binding sites between Nrf2 and Keap1 (DLG-motif) and hence protects Nrf2 from Keap1/Cul3-mediated degradation, resulting in a stabilization of Nrf2 protein levels[6]. Hence, treatment with SFN in this cell line is not expected to increase Nrf2 levels (as seen in Supplementary Fig. 1b), but compound 7 is expected to block binding of Keap1 to Nrf2 that occurs via the still intact second binding site (ETGE-motif). As expected, while treatment with compound 7 decreased the signal in both cell lines in a dose-dependent manner, treatment with SFN had no effect in the cell line expressing Nrf2-D29H while provoking a dose-dependent increase in the cell line expressing wild-type Nrf2. Taken together, our data introduces a method to study protein-protein interactions in cells in real-time over an extended amount of time. Hence we named the method real-time-bind (RT-Bind).

**EndoBind detects protein–protein interactions of endogenous proteins in real-time.** While RT-Bind offers the advantage of measuring the interaction of proteins of interest kinetically over an extended period and employs a standardized expression promoter permitting normalization for transcriptional effects, it still relies on ectopic expression of fusion proteins, making RT-Bind prone to artifacts due to non-physiological interaction of the partners. To circumvent this liability, we wondered whether we could measure the interaction of endogenous proteins at physiological expression levels by introducing the NanoBit-tags at the respective genomic loci using CRISPR-knockin instead of overexpressing the respective fusion proteins. To test this particular hypothesis, we set out to measure the interaction of G12V-mutant KRAS and CRAF, two proteins that signal via forming complexes with each other and other proteins. Artificial overexpression of RAS can alter its physiological interactions and signaling behavior in cells, leading to increased cell growth, senescence, or apoptosis[9–11] and making an endogenous tagging approach highly desirable. We decided to use the pancreatic cell line PATU8988T for three reasons: (1) it contains a homozygous G12V mutation in KRAS (i.e., one of the most common KRAS-activating mutations in cancer[12]), (2) the KRAS locus is not amplified, (3) its high transfection efficiency. We sequentially introduced the SmB at the N-terminus of KRAS and the LgB at the N-terminus of CRAF via CRISPR-knockin in PATU8988T cells and selected a single-cell clone featuring homozygous tagging at both loci for further studies (Supplementary Fig. 2a). The introduction of the tags shifted the size of the endogenous proteins as expected while exerting no discernable effect on endogenous expression levels, localization of the tagged proteins, and growth rate of the cells (Supplementary Fig. 2b, c). Using a similar workflow to RT-Bind we were able to detect a low but significant basal signal in the PATU8988T-SmBKRAS-LgBCRAF cell line. We termed the use of the RT-Bind workflow with endogenously tagged partners, EndoBind.

To further test the limits of the KRAS:CRAF EndoBind assay, we treated the cells with the type-1 RAF-inhibitor GDC0879 which is expected to increase the interaction of KRAS and CRAF[13]. Treatment of the cells with GDC0879 indeed yielded a dose-dependent increase in signal in PATU8988T-SmBKRAS-LgBCRAF cells but not in the isogenic parental cells (Fig. 2a). This signal increase was most pronounced 7 h after treatment and was still detectable 24 h after treatment. To demonstrate signal dependency on the specific interaction of KRAS and CRAF, we transfected the cells with siRNAs targeting either CRAF or KRAS and then treated the cells with GDC0879 for 7 h (Fig. 2b). As expected, the EndoBind signal depended on the expression of both KRAS and CRAF and showed a concentration-dependent decrease after siRNA-mediated knockdown of either CRAF or KRAS that correlated with the remaining protein levels. There was no significant decrease in EndoBind signal after knockdown of CRAF or KRAS in the absence of GDC0879, indicating that there is no detectable interaction of CRAF and KRAS in the absence of GDC0879 (Supplementary Fig. 2d).

To test the extensibility of this EndoBind assay we treated the cells with either GDC0879 (type 1 RAF-inhibitor), LHS533 (type 1.5 RAF-inhibitor), LY3009120 (type 2 RAF-inhibitor)[13], or Selumetinib (MEK-inhibitor)[14] and measured the KRAS:CRAF

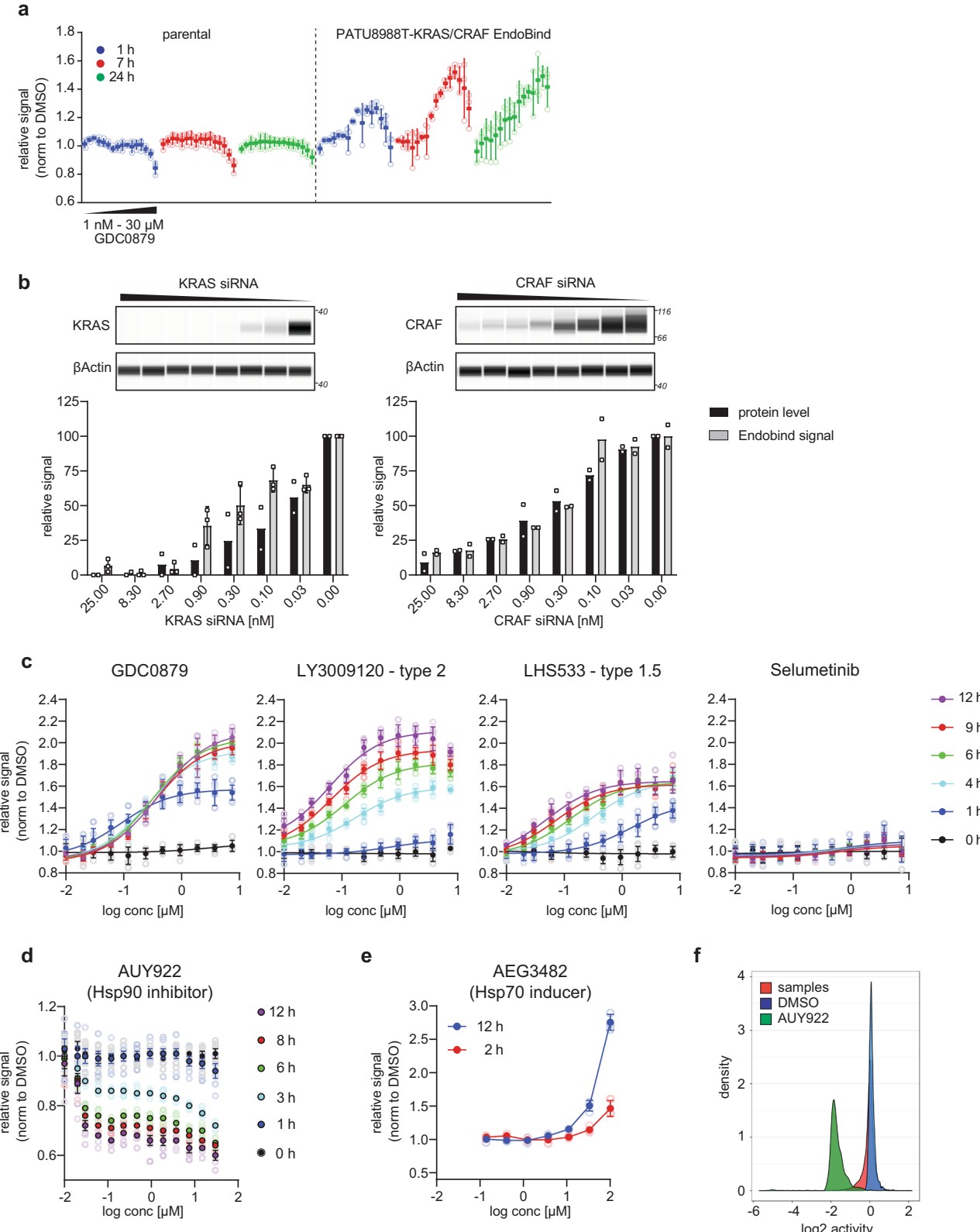

interaction after the indicated duration of treatment (Fig. 2c). The binding of RAF inhibitors to RAF in the presence of activated KRAS leads to allosteric priming, in which the RAF inhibitor promotes interaction of RAF with activated RAS by stabilizing the activated conformation of RAF[15]. According to this model, RAF inhibitors that bind to both protomers in a dimer (type 1 and type 2) show stronger allosteric priming than inhibitors of monomeric RAF (type 1.5). Coherent with this model, all three RAF-inhibitors led to an increase in EndoBind signal, indicative of increased interaction of KRAS and CRAF. As expected, the type 1.5 inhibitor LHS533 showed the lowest induction of EndoBind signal, while both GDC0879 and LY3009120 provoked much stronger increases in signal, albeit with different kinetics, peaking at 4 h compared to 12 h post treatment, respectively.

**Fig. 2 EndoBind allows for the detection of the interaction of endogenous KRAS and CRAF in PATU8988T cells. a** EndoBind signal in PATU8988T-SmBKRAS-LgBCRAF cells or parental PATU8988T cells treated with GDC0879 for 1, 7, or 24 h (mean ± s.d., $n = 3$ wells each, representative experiment). Data were normalized to the control treatment (DMSO). **b** EndoBind signal after knockdown of KRAS or CRAF in PATU8988T-SmBKRAS-LgBCRAF cells and treatment with GDC0879 (mean ± s.d., $n = 3$ for siKRAS, mean of $n = 2$ for siCRAF). Signals were normalized to cells transfected with a non-targeting siRNA. Protein levels after transfection of the indicated amount of siRNA were quantified using the Jess system (mean, $n = 2$). **c** EndoBind signal in PATU8988T-SmBKRAS-LgBCRAF cells after treatment with the RAF-inhibitors GDC0879, LY3009120, NVP-LHS533, and the MEK-inhibitor Selumetinib. The same plate was measured at the indicated time points (mean ± s.d., $n = 3$ wells each, representative experiment). Data were normalized to the control (DMSO) treated sample. **d** EndoBind signal in PATU8988T-SmBKRAS-LgBCRAF cells after treatment with the Hsp90 inhibitor NVP-AUY922 and GDC0879 (mean ± s.d., $n = 3$ wells each, representative experiment). Data were normalized to the control (DMSO) treated sample. **e** EndoBind signal in PATU8988T-SmBKRAS-LgBCRAF cells after treatment with the Hsp70 inducer AEG3482 and GDC0879 (mean ± s.d., $n = 3$ wells each, representative experiment). Data were normalized to the control (DMSO) treated sample. **f** Distribution of EndoBind signal across the 35 plates (1536-well plates) of a 50 k compound screen in PATU8988T-SmBKRAS-LgBCRAF cells after treatment with NVP-AUY922 (active control), DMSO (neutral control), or sample compound in the presence of GDC0879. Data were normalized to the neutral control (DMSO) on each plate.

To demonstrate that the EndoBind assay is suited to identifying compounds with the mechanism of action other than RAF inhibition, we treated the cells with either an Hsp90 inhibitor or an Hsp70 inducer. Both chaperones are required for the stability and activity of CRAF[16,17]. As expected, the Hsp90 inhibitor NVP-AUY922[18] led to time- and dose-dependent decreases in EndoBind signal likely due to degradation of CRAF protein (Fig. 2d). Similarly, AEG3482, an Hsp70 inducer[19], led to an increase in EndoBind signal due to facilitating CRAF protein folding and stability (Fig. 2e).

Lastly, we demonstrated the suitability of the EndoBind assay for running a fully automated 50 k well screen in 1536-well format. Figure 2f shows the distribution of the signal across the screening plates, demonstrating a good separation of signal between the neutral control DMSO and the active control NVP-AUY922.

Taken together, we describe two new assays formats that allow for the continuous detection of protein-protein interactions in real-time for ectopically expressed or endogenously tagged proteins in cells. The use of a pro-substrate that can be added during seeding of the cells eliminates the need for additional substrate addition at the time of measurement and enables high-throughput-friendly, continuous readout of the signal on the same plate over an extended amount of time as well as pre- and post-treatment measurement on the same cells.

## Methods

**Plasmids**. Plasmids for RT-Bind: The coding sequences for SmB and LgB, flanked by a linker region with additional restriction sites, were cloned into a pCDH1-EF1 vector (System Biosciences), generating a set of 8 unique vectors with either N-terminal or C-terminal NanoBit-tags and either a Hygromycin- or a Puromycin-cassette driven by a CMV-promoter. The restriction sites were designed as such that the same ORF-sequence (without stop codon) can be cloned in all eight vectors. Sequences for Nrf2 were cloned via the NotI/SbfI-sites and were amplified using the following primer sequences: NRF2-F: TCCTCCGCGGCCGCATGAT GGACTTGGAGCTGCC, NRF2-R: TCCTCCCCTGCAGGGTTTTTCTTAACAT CTGGCTTC. Sequences for Keap1 were cloned via the NotI/EcoRV-sites and were amplified using the following primer sequences: KEAP1-F: TCCTCCGCGGCCG CATGCAGCCAGATCCCAGGCC, KEAP1-R: TCCTCCGATATCACAGGTAC AGTTCTGCTGGTCAAT. Final vector sequences were validated by sequencing.

Plasmids for CRISPR-knockin: We designed a donor-plasmid with the sequence for either SmB or LgB and a V5- or FLAG-tag in frame with an upstream sequence of Blasticidin or Puromycin, separated by a P2A site, flanked on each side with 800bp complementary sequence upstream and downstream of the integration site (Supplementary Fig. 2a). In-frame integration of the template sequence at the N-terminus of KRAS or CRAF, respectively, results in the expression of a Blasticidin- or Puromycin-resistance cassette and the N-terminally tagged protein separated by a P2A-site. The donor vector also contained an eGFP-sequence driven by an EIF1a-promotor outside the homology arms to enable negative selection for cells with unspecific integration. The respective sequence was synthesized and cloned into a puc57-vector. All gene synthesis and cloning were performed by Genscript. Full sequences of the inserts can be found in the source data. The sgRNAs targeting the integration sites were cloned into a modified U6-based puromycin-resistant pLKO vector, including an expression cassette for *Streptococcus pyogenes* Cas9 with an N-terminal SV40 nuclear localization signal and a C-terminal SV40 nuclear localization signal driven by a CMV promoter.

sgRNA-KRAS: AATGACTGAATATAAACTTGTGG, sgRNA-CRAF: GCATCA ATGGAGCACATACA.

**Cell culture**. HEK293T and PATU8988T cells were cultured in DMEM supplemented with 10% fetal bovine serum (FBS) and 1% penicillin–streptomycin (all from Life Technologies) at 37°C with 5% $CO_2$.

**Virus packaging protocol**. Lentivirus was generated in a 96-well format as described here[20]. Briefly, $4 \times 10^4$ HEK293T cells were transfected with 83 ng pMLDg/pRRE, 32 ng pRSC-Rev, 45 ng pVSV-G, and 100 ng construct of interest using FUGENE 6. Twenty-four-hour after transfection medium was changed and the virus was collected 48 h after transfection. The virus was frozen prior to transduction to prevent carry-over of HEK293T cells.

**Generation of stable cell lines for RT-Bind assay**. HEK293T cells were seeded in 6-wells and transduced with a low multiplicity of infection with lentivirus containing the respective constructs. Cells expressing the LgB-tagged protein of interest were selected with 1.75 µg/ml Puromycin, cells expressing the SmB-tagged protein of interest were selected with 250 µg/ml Hygromycin, cells expressing both constructs were selected with both.

**Generation of CRISPR-knockin cell line**. PATU8988T-SmB-KRAS cell lines were generated by co-transfecting 300 ng of the donor template with 300 ng of the sgRNA- and Cas9-encoding vector using 1.8 µl Dharmafect kb (Dharmacon) per 12-well. Transfected cells were cultured and expanded for 5 days before 30 µg/ml Blasticidin was added. After the outgrowth of a Blasticidin-resistant population of cells, the cells were sorted on an Aria-FACS-Sorter (BD Bioscience) and the 10% most GFP-negative population was selected. Cells were single-cell-cloned by serial dilution and clones were analyzed by PCR and western blotting for the successful integration of the donor sequence. Once a clone with homozygous integration was identified, the same process was repeated with the donor construct for N-terminal tagging of CRAF, using Puromycin selection (1 µg/ml).

**Transfection with siRNA**. siRNA targeting KRAS (Dhamarcon, cat No. L-005069-00-0005) or CRAF (Dhamarcon, cat No. L-003601-00-0005) was diluted using control siRNA (Dhamarcon, cat No. D-001810-10-05). The transfection mixture was prepared by mixing siRNA with 0.187 µl of Lipofectamine RNAiMAX Transfection Reagent (Thermo Fisher, cat No. 12778100) in 20 µl of Opti-MEM Reduced Serum Medium (Thermo Fisher, cat No. 31985062). The mixture was incubated at room temperature for 30 min before adding 20 µl of cell suspension containing $4 \times 10^3$ cells in a 384-well white solid bottom assay plate. EndoBind signal was determined 48 h post siRNA transfection.

A similar transfection protocol was used for siRNA transfection for WES. Briefly, siRNA prepared with 2.16 µl of RNAiMax in 320 µl of Opti-MEM medium was added into an equal volume of cell suspension containing at $2 \times 10^5$ cells/ml in a 24-well plate. Forty-eight hours post siRNA transfection, the cells were subjected to WES western blot analysis.

**Western blotting**. Forty-eight hours post siRNA transfection, the cells were lysed in RIPA buffer (Cell Signaling, cat No. 9806) and subjected to WES western blot analysis using antibodies for KRAS (Life Technologies, cat No. 415700), CRAF (Abcam, cat No. b137435) or β-Actin (Cell Signaling Technology, cat No. 3700) on a WES Separation Module (Protein Simple, cat No. SM-W004) according to the protocol suggested by the manufacturer. For regular western blot analysis, cells were lysed in RIPA buffer and 20 µg total lysate was subjected to sodium dodecyl sulfate-polyacrylamide gel electrophoresis using NuPAGE 4–12% Bis/Tris gels (ThermoFisher, cat No. NP0321). Subcellular fractionation was done using the Mem-PER™ Membrane Protein Extraction Kit (ThermoFisher, cat No. 89842). Proteins were transferred to nitrocellulose membranes (ThermoFisher, cat No.

LC2000) by wet transfer. The membrane was blocked with Intercept (TBS) Blocking buffer (LICOR, cat No. 927-60001), incubated with antibodies directed against pan-RAS (EMD Millipore, cat No. 050516), KRAS (Abnova, cat No. H00003845-M01), CRAF (Cell Signaling Technology, cat No. 53745), NRF2 (Abcam, cat No. ab62352), Keap1 (Cell Signaling Technology, cat No. 8047), pan-Cadherin (Cell Signaling Technology, cat no. 4068), GAPDH (Cell Signaling Technology, cat no. 97166), Hsc70 (Enzo Life Sciences, cat No. ADI-SPA-820-F), FLAG (Sigma/Millipore, cat No. F1804), V5 (ThermoFisher, cat No. R96025), and subsequently with secondary antibodies (IRDye 800CW anti-rabbit, LICOR, cat No. 926-32213 and IRDye 680RD anti-mouse, LICOR, cat No. 926-68072).

**Luciferase measurements**. Cells were cultured in the appropriate size of the flask to ensure confluence at the time of harvest was <80%. Cells were harvested and diluted in DMEM + 10% FBS to a final concentration of $0.3 \times 10^6$ cells/ml (for 384-well format) or $1 \times 10^6$ cells/ml (for 1536-well format). RealTime-Glo substrate (Promega, cat No. G9711) was added at a dilution of 1:2000. Thirty microlitres of cells were plated per 384-well (10,000 cells/well) and 5 µl of cells were plated per 1536-well. Cells were incubated overnight. Baseline luminescence was measured the next morning before the cells were treated as described. For RT-Bind and EndoBind luminescence was measured using the Viewlux system (PerkinElmer) or the luminescence plate reader (LPR, GNF) with a 2 min exposure. For all other luminescence measurements, NanoGlo (Promega, cat No. N1110) or NanoGlo-live (Promega, cat No. N2011) was added according to the manufacturer's protocol and luminescence was measured as described.

**Compound treatment**. Compounds were synthesized in-house. Totally, 10 mM stock solutions were made in DMSO. Compounds were dosed using the Echo acoustic liquid handling technology (Labcyte).

**Screen**. The compound library was prespotted in 1536-well plates and 5 µl of a suspension of PATU8988T-SmBKRAS-LgBCRAF cells mixed with RealTime-Glo substrate (Promega, cat No. G9711) were added. Twenty-four-hour after plating, the baseline EndoBind-signal was measured on the luminescence plate reader (LPR, GNF). The cells were treated with 5 µM GDC0879 for 1 h and the EndoBind-signal was measured again. Three micrometer NVP-AUY922 was used as a control.

**Statistics and reproducibility**. All quantitative data are expressed as mean ± standard deviation. The number and nature of replicates (*n*) are given in each figure legend. In case representative data of one experiment is shown, the experiment was repeated at least one additional time with a similar outcome. Data analysis and graphing were performed with Prism 9 (GraphPad software).

**Reporting summary**. Further information on research design is available in the Nature Research Reporting Summary linked to this article.

## Data availability
All data shown or analyzed during this study are included in this article and its supplementary information file. Uncropped western blot images can be found in the Supplementary Data file.

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

## Author contributions
A.B., D.G., J.R.H., L.M., T.P.O., and F.K. designed the experiments and prepared the paper, A.B., S.E., M.L., M.Lu., A.Ma., A.Me., B.T., and L.V. performed the experiments and analyzed the results. All authors approved the paper.

## Competing interests
The authors declare no competing interests.
