## [Peer Review File · Communications Biology]

Reviewers' comments:

Reviewer #1 (Remarks to the Author):

Brief summary of the manuscript

In their manuscript, Bill, A. et al. present two approaches enabling detection and monitoring of protein-protein interactions (PPIs) upon their ectopic (RT-Bind) and endogenous (EndoBind) expression. Both methods allow to continuously monitor PPIs in real time and over an extended duration. For the RT-Bind approach the authors employed an interaction between KEAP1 and NRF2 and developed a stable HEK293T cell line overexpressing both proteins tagged with the split luciferase subunits. Next, the authors used some known modulators of KEAP1/NRF2 interaction to demonstrate the suitability of this approach. In the second part of the study the authors introduced the split luciferase subunits and additional tags upstream endogenous sequences encoding for KRAS and CRAF (for this purpose a pancreatic cell line PATU8988T has been employed) and again used some inducers and modulators of the interaction between these two proteins to verify the EndoBind approach they had developed. The authors also examined how targeting either KRAS or CRAF with siRNA will affect their interaction.

Overall impression of the manuscript

The approaches presented in the manuscript are novel. The article is well-written, well-documented, well-presented and, with few exceptions, easy to follow. The research design is appropriate. The conclusions are fully supported by the rigorous, diverse and complementary experimental data. The results are convincing. The research methodology is comprehensively described so that the experiments can be easily reproduced. The strong points of the study include generation of stable transfectants, monitoring protein expression levels, using RNAi strategy and a particularly innovative approach involving CRISPR/Cas9-assisted tagging of endogenously expressed proteins with split luciferase subunits. I predict that the impact of the article on the field will be high as the RT-Bind and EndoBind approaches developed by the authors can be employed in a variety of studies and therefore should be of great interest to the scientific community. Moreover, the article is highly timely as the luminescence-based approaches are gaining more and more attention.

Major comments:

The work is sound but requires some improvements. I would ask for clarification of the following issues and recommend the following amendments:

- 1) Although the NanoBiT strategy is relatively novel, it has been employed in many studies so far. Could the authors reference 2-3 more articles that utilized this approach?
- 2) Tagging of proteins may alter their subcellular localization. Did the authors check, whether the staining pattern of KRAS and CRAF proteins tagged with the split luciferase subunits and additional epitopes is the same as that of the unmodified proteins? Providing images obtained via immunofluorescence staining using antibodies mentioned in the Methods section could greatly improve the corresponding part of the manuscript.
- 3) Could the authors substantiate the choice of the cell lines used?
- 4) Could the authors explain if there is any difference between KRAS and KRAS-G12V? Are these terms used interchangeably?
- 5) Could the authors explain, why there are two bands visible in Supplementary Figure 2B (the blot referring to KRAS) in the lane corresponding to the engineered cells?
- 6) Could the authors improve the quality of the pictures by increasing the size of the plots and/or the font size? Some of them (e.g. Fig. 2B, 2C and 2G) are very difficult to decipher.
- 7) Could the authors provide a little more explanation and a reference to the statement claimed in lines 106-109?
- 8) For the sake of clarity, could the authors provide an extended description of the transfectants and/or combinations of the expression constructs, e.g. HEK293T-SmB-NRF2/Keap1-LgB; HEK293T cells co-expressing NRF2 N-terminally tagged with SmB and Keap1 C-terminally tagged with LgB, somewhere in the text?

Minor comments:

1. In line 16 an abbreviation "KRAS_G12V" is used, whereas in the rest of the manuscript (e.g. line 107) it is "KRAS-G12V", please unify.
2. In line 32 a term „Nano-luciferase" is used, whereas in the rest of the manuscript this enzyme is being referred to as NanoLuciferase, please unify.
3. Line 34: there should be "furimazine" instead of "fumirazine".
4. I do not quite understand why NRF2 is written in capital letters, whereas Keap1 is not, should not these abbreviations be written in a uniform manner?
5. Line 41: please explain abbreviations "SmB" and "LgB" in the text and unify them throughout the text (e.g. line 56).
6. Line 45: a term "promotor" is used, whereas in the other fragments of the manuscript (e.g. line 94) there is "promoter", please unify.
7. A reference to Supplementary Figure 1 should be given earlier in the text, preferably in line 48.
8. A style used in the caption of Supplementary Figure 1 is inconsistent (A is followed by a full stop, whereas B and C are followed by commas) - please unify (this refers to other figure captions as well).
9. Line 49: please explain abbreviation "Cul-3" earlier in the text and unify the corresponding abbreviation between lines 49, 74 and 94.
10. Line 65: there should be "subsequently" instead of "subsequent".
11. Line 117: there is a double space between words "of" and "the".
12. Line 146: "both" and "and" should not be written in italics.
13. Line 151: there should be "EndoBind" instead of "Endobind".
14. Line 199: there should be "an EF1a" instead of "a EIF1a".
15. Lines 204-205: "Streptococcus pyogenes" should be written in italics.
16. Line 209: the degree symbol is missing.
17. Lines 211, 236, 240 - please use a consistent way of giving cell numbers.
18. Please use a consistent manner of giving units (sometimes a unit is separated from a numerical value by a space and sometimes it is not).
19. Please use the "μ" symbol instead of "u" to denote the "micro" unit prefix throughout the text.
20. Line 228: there should be "Puromycin selection" instead of "Puromycin-selection".
21. Line 234: there should be "20 μl" instead of "20 up".
22. Lines 245, 251 and 252: there should be "Cell Signalling Technology" instead of "Cell Signalling".
23. Line 246: there should be e.g. "protocol suggested by the manufacturer" instead of "protocol by the manufacturer".
24. Line 269: there should be "solutions" instead of "solution".
25. Caption of Supplementary Figure 2, line 21: please modify the description as the cells were not treated with GDC.

Reviewer #2 (Remarks to the Author):

I have read "EndoBind – Real-time detection of endogenous protein-protein interactions," by Dr. Bill et al. This work aims to apply the commercially available NanoLuciferase system to measure the NRF2/KEAP1 interaction and the as well as a KRAS(V12)-CRAF interaction using a lentiviral integration or CRISPR mediated insertion into the endogenous gene loci. They demonstrate the commercial applicability for compound screening by performing measurements in 384 and 1536-well plates.

Overall, the paper provides a useful demonstration of the use of the NanoLuc system to two practical examples. The data are clear and the experiments presented should be replicable. Editing is needed to improve the readability of the paper. Often the language uses vague reference, leaving the reader guessing at the meaning. For example, page 2, line 25 "Analyzing the status-quo at a specific time..." I have no idea what this is referring to. A more specific introduction of the problem and the approach would help. Moreover, no introduction or citation is given for smBit or IgBit, results page 5. While these are known by some, rigor in the introduction and citation

would also help make the work more clear and accessible.

Please provide a reference and explanation for the use of NRF2-D29H mutation. A more cogent explanation for its effects in the results section would make this more readable.

Reviewer #3 (Remarks to the Author):

In this manuscript, the King laboratory optimizes a technique designed to detect the interaction of ectopically expressed (RT-Bind) or endogenously tagged (EndoBind) proteins of interest. The authors' methods basically follow the manufacturer's (Promega Corporation) instruction (NanoBiT® Protein:Protein Interaction System) except for the use of pro-substrate, which used in other assay kit in this company, RealTime-Glo. This optimization is useful, however, the optimization does not use particularly novel technologies, but instead simply employs existing methods or materials. Moreover, although the advantage of using the pro-substrate is the detection of signal for longer time (pro-substrate: over 2 days), certainly fumirazine has a short half-life of about 2 hours, but other substrates provided for this interaction assay by Promega, Vivazine or Endurazine, have longer half-lives (about 10-20 hours or several days, respectively). Therefore, it does not appear to be novel unless there are dramatic advantages over these substrates.

Point-by-point response to the Reviewers' comments:

We thank the reviewers for taking the time to thoroughly review our manuscript and for their comments and suggestions to improve our manuscript. We have addressed the specific points raised by the Reviewers by adding additional data as well as any necessary corrections, clarification and/or discussion.

Reviewer #1 (Remarks to the Author):

Brief summary of the manuscript

In their manuscript, Bill, A. et al. present two approaches enabling detection and monitoring of protein-protein interactions (PPIs) upon their ectopic (RT-Bind) and endogenous (EndoBind) expression. Both methods allow to continuously monitor PPIs in real time and over an extended duration. For the RT-Bind approach the authors employed an interaction between KEAP1 and NRF2 and developed a stable HEK293T cell line overexpressing both proteins tagged with the split luciferase subunits. Next, the authors used some known modulators of KEAP1/NRF2 interaction to demonstrate the suitability of this approach. In the second part of the study the authors introduced the split luciferase subunits and additional tags upstream endogenous sequences encoding for KRAS and CRAF (for this purpose a pancreatic cell line PATU8988T has been employed) and again used some inducers and modulators of the interaction between these two proteins to verify the EndoBind approach they had developed. The authors also examined how targeting either KRAS or CRAF with siRNA will affect their interaction.

Overall impression of the manuscript

The approaches presented in the manuscript are novel. The article is well-written, well-documented, well-presented and, with few exceptions, easy to follow. The research design is appropriate. The conclusions are fully supported by the rigorous, diverse and complementary experimental data. The results are convincing. The research methodology is comprehensively described so that the experiments can be easily reproduced. The strong points of the study include generation of stable transfectants, monitoring protein expression levels, using RNAi strategy and a particularly innovative approach involving CRISPR/Cas9-assisted tagging of endogenously expressed proteins with split luciferase subunits. I predict that the impact of the article on the field will be high as the RT-Bind and EndoBind approaches developed by the authors can be employed in a variety of studies and therefore should be of great interest to the scientific community. Moreover, the article is highly timely as the luminescence-based approaches are gaining more and more attention.

Response: We thank the Reviewer for pointing out the novelty of our approaches and the rigorous experimental design and data. We agree with the Reviewer that the presented techniques will be of high interest to the scientific community as they can be easily employed for multiple studies using standard laboratory equipment.

Reviewer #1 (Major comments):

The work is sound but requires some improvements. I would ask for clarification of the following issues and recommend the following amendments:

1) Although the NanoBiT strategy is relatively novel, it has been employed in many studies so far. Could the authors reference 2-3 more articles that utilized this approach?

Response: We apologize for the mishap and we added additional references to other studies using the NanoBiT technology. The manuscript was updated as follows: "Despite its only recent publication, the NanoBiT™ system has been successfully used to study protein:protein interactions of cytoplasmic and membrane proteins at a given timepoint (Bodle, et al 2017, Grane-Boladeras et al, 2019, Leroy et al, 2019, Storme et al, 2018)." (Page 2, lines 38-40)

2) Tagging of proteins may alter their subcellular localization. Did the authors check, whether the staining pattern of KRAS and CRAF proteins tagged with the split luciferase subunits and additional epitopes is the same as that of the unmodified proteins? Providing images obtained via immunofluorescence staining using antibodies mentioned in the Methods section could greatly improve the corresponding part of the manuscript.

Response: We thank the Reviewer for this constructive comment. We performed a cellular fractionation experiment demonstrating that the tagged proteins in the PATU8988T-SmBKRAS-LgBCRAF cell line show the same localization as in the parental cell line. The data was added as Supplementary Figure 2C (see below) and the text in the manuscript has been updated as follows: “Introduction of the tags shifted the size of the endogenous proteins as expected while exerting no discernable effect on endogenous expression levels, localization of the tagged proteins and growth rate of the cells (Supplementary Figure 2B, C)”. (Page 5, lines 111-114) Please note, that to our knowledge, there are currently no suitable antibodies for detecting KRAS via immunofluorescence imaging.

3) Could the authors substantiate the choice of the cell lines used?

Response: We added additional reasoning about the choice of cell lines used as follows: “HEK293T cells do not harbor mutations in Nrf2 and Keap1 and show good transduction efficiencies; hence they were a suitable cell line for our study.” (Page 2, lines 48-50) and “We decided to use the pancreatic cell line PATU8988T for three reasons: 1) it contains a homozygous G12V mutation in KRAS (i.e. one of the most common KRAS-activating mutation in cancer¹²), 2) the KRAS locus is not amplified, 3) its high transfection efficiency.” (Page 4, line 106 – page 5 line 108).

4) Could the authors explain if there is any difference between KRAS and KRAS-G12V? Are these terms used interchangeably?

Response: We apologize for the confusion and thank the Reviewer for pointing it out. We updated the manuscript to make it more clear that we are referring to KRAS as the general protein and we pointed out the G12V mutation when introducing the cell line used. (see previous comment)

5) Could the authors explain, why there are two bands visible in Supplementary Figure 2B (the blot referring to KRAS) in the lane corresponding to the engineered cells?

Response: We thank the Reviewer for making us aware of this potentially confusing observation. The antibody used for the western blot in Supplementary Figure 2B is a pan-RAS antibody. The lower band

corresponds to HRAS and NRAS, while KRAS runs slightly higher (there is a double band in the parental cell line). In the PATU8988T-SmBKRAS-LgBCRAF cell line, only the band for KRAS shifts up (because of the addition of the SmB), demonstrating the specificity of the tagging approach. We confirmed this observation by using a KRAS-specific (instead of pan-RAS) antibody in Supplementary Figure 2C. To avoid confusion, we added arrows indicating the expected size of the untagged/tagged proteins and additional explanation as well as a reference to the validation data for the pan-RAS antibody to the figure legend. *“Note: The pan-RAS antibody detects all RAS forms. The lower band corresponds to H- and NRAS, while the upper band corresponds to KRAS (Waters et al., 2017). Not the upshift of the KRAS-band after addition of the tag.”* (Page 2, lines 19-21, Supplementary Figures)

6) Could the authors improve the quality of the pictures by increasing the size of the plots and/or the font size? Some of them (e.g. Fig. 2B, 2C and 2G) are very difficult to decipher.

Response: We apologize for the bad image quality. We transformed all figures to vector based graphics and include them as separate pdfs instead of inserting them into the manuscript text.

7) Could the authors provide a little more explanation and a reference to the statement claimed in lines 106-109?

Response: We updated the manuscript to include more rationale why overexpression of KRAS can be problematic and why an endogenous tagging strategy can be beneficial. *“To test this particular hypothesis, we set out to measure the interaction of G12V-mutant KRAS and CRAF, two proteins that signal via forming complexes with each other and other proteins. Artificial overexpression of RAS can alter its physiological interactions and signaling behavior in cells, leading to increased cell growth, senescence or apoptosis (Birchler et al, 2012, Narita et al, 2005, Vartanian et al, 2013) and making an endogenous tagging approach highly desirable”.* (Page 4, lines 102-106)

8) For the sake of clarity, could the authors provide an extended description of the transfectants and/or combinations of the expression constructs, e.g. HEK293T-SmB-NRF2/Keap1-LgB; HEK293T cells co-expressing NRF2 N-terminally tagged with SmB and Keap1 C-terminally tagged with LgB, somewhere in the text?

Response: We apologize for not providing enough description of the tagging in the cell lines used. We included additional information in the text to clearly indicate which tag position was used. *“The cell line with N-terminal SmB-tag on Nrf2 and C-terminal LgB-tag on Keap1 (from herewith referred to as SmB-Nrf2/Keap1-LgB) was chosen for all further experiments...”* (Page 3, lines 54-55) and *“We sequentially introduced the SmBit at the N-terminus of KRAS and the LgBit at the N-terminus of CRAF via CRISPR-knockin in PATU8988T cells”* (Page 5, lines 108-110).

Reviewer #1 (Minor comments):

1. In line 16 an abbreviation “KRAS_G12V” is used, whereas in the rest of the manuscript (e.g. line 107) it is “KRAS-G12V”, please unify.
2. In line 32 a term „Nano-luciferase” is used, whereas in the rest of the manuscript this enzyme is being referred to as NanoLuciferase, please unify.
3. Line 34: there should be “furimazine” instead of “fumirazine”.
4. I do not quite understand why NRF2 is written in capital letters, whereas Keap1 is not, should not these abbreviations be written in a uniform manner?

Response: We thank the Reviewer for making us aware of these inconsistencies and we corrected the spelling and ensured uniformity of the respective abbreviations throughout the manuscript.

5. Line 41: please explain abbreviations “SmB” and “LgB” in the text and unify them throughout the text (e.g. line 56).

Response: We added more explanation to the text when first mentioning the abbreviations “SmB” and “LgB” and corrected to manuscript text and figure legends to use consistent abbreviations. *“For this, the proteins are reciprocally fused to one of the two optimized portions of NanoLuciferase (the smaller part, SmB; or the larger part, LgB) via tailored, flexible linkers.”* (Page 2, lines 36-37)

6. Line 45: a term “promotor” is used, whereas in the other fragments of the manuscript (e.g. line 94) there is “promoter”, please unify.

Response: We corrected the spelling of the word promoter throughout the manuscript.

7. A reference to Supplementary Figure 1 should be given earlier in the text, preferably in line 48.

Response: We now reference Supplementary figure in line 57 (previous line 48).

8. A style used in the caption of Supplementary Figure 1 is inconsistent (A is followed by a full stop, whereas B and C are followed by commas) - please unify (this refers to other figure captions as well).

Response: We unified the format of all figure legends.

9. Line 49: please explain abbreviation “Cul-3” earlier in the text and unify the corresponding abbreviation between lines 49, 74 and 94.

Response: Cullin3 (Cul3) is now introduced in line 48 and we unified the abbreviation in the text.

10. Line 65: there should be “subsequently” instead of “subsequent”.
11. Line 117: there is a double space between words “of” and “the”.
12. Line 146: “both” and “and” should not be written in italics.
13. Line 151: there should be “EndoBind” instead of “Endobind”.
14. Line 199: there should be “an EF1a” instead of “a EIF1a”.
15. Lines 204-205: “Streptococcus pyogenes” should be written in italics.
16. Line 209: the degree symbol is missing.
17. Lines 211, 236, 240 - please use a consistent way of giving cell numbers.
18. Please use a consistent manner of giving units (sometimes a unit is separated from a numerical value by a space and sometimes it is not).
19. Please use the “ μ ” symbol instead of “u” to denote the “micro” unit prefix throughout the text.
20. Line 228: there should be “Puromycin selection” instead of “Puromycin-selection”.
21. Line 234: there should be “20 μ l” instead of “20 up”.
22. Lines 245, 251 and 252: there should be “Cell Signalling Technology” instead of “Cell Signalling”.
23. Line 246: there should be e.g. “protocol suggested by the manufacturer” instead of “protocol by the manufacturer”.
24. Line 269: there should be “solutions” instead of “solution”.
25. Caption of Supplementary Figure 2, line 21: please modify the description as the cells were not treated with GDC.

Response: We thank the Reviewer for pointing out all these inconsistencies and typos. We made the respective corrections throughout the text and the figure legends.

Reviewer #2 (Remarks to the Author):

I have read “EndoBind – Real-time detection of endogenous protein-protein interactions,” by Dr. Bill et al. This work aims to apply the commercially available NanoLuciferase system to measure the NRF2/KEAP1 interaction and the as well as a KRAS(V12)-CRAF interaction using a lentiviral integration or CRISPR mediated insertion into the endogenous gene loci. They demonstrate the commercial applicability for compound screening by performing measurements in 348 and 1536-well plates.

Overall, the paper provides a useful demonstration of the use of the NanoLuc system to two practical examples. The data are clear and the experiments presented should be replicable. Editing is needed to improve the readability of the paper. Often the language uses vague reference, leaving the reader guessing at the meaning.

Response: We thank the Reviewer for reviewing our manuscript and for providing constructive feedback. We thank the Reviewer for highlighting that we provided practical examples and that the data are clear and replicable.

#1: For example, page 2, line 25 “Analyzing the status-quo at a specific time...” I have no idea what this is refereeing to. A more specific introduction of the problem and the approach would help.

Response: We apologize for not being more precise in our introduction. We rephrased the first paragraph of the text to provide a better introduction to the limitation of current methodology: “*Cell signaling is a highly flexible and transient process that relies on a careful balance between inhibitory and activating signals mediated by protein-protein interactions. A variety of methods for monitoring cellular protein-protein interactions have been described that suffer enormous limitations to capture those highly dynamic signaling events in cells; i) Ectopic, usually transient overexpression risks non-biologically relevant interactions or artificial modulation of the signaling event itself by disturbing the stoichiometric balance. ii) Analyzing the protein-protein interactions or signaling state of a cell at a specific time point in end-point assays can lead to misinterpretation or oversight of important underlying processes. And iii) Methods relying on the detection of endogenous protein require either high amounts of protein or special instrumentation, making them incompatible with high-throughput applications.*” (Page 1 line 21 - page 2 line 29)

#2: Moreover, no introduction or citation is given for smBit or IgBit, results page 5. While these are known by some, rigor in the introduction and citation would also help make the work more clear and accessible.

Response: We updated the introduction of the paper to include a more detailed description of the NanoBit technology, including more references and specifically introduce SmB and LgB: “*For this, the proteins are reciprocally fused to one of the two optimized portions of NanoLuciferase (the smaller part, SmB; or the larger part, LgB) via tailored, flexible linkers.*” (page 2, lines 36-37)

#3: Please provide a reference and explanation for the use of NRF2-D29H mutation. A more cogent explanation for its effects in the results section would make this more readable.

Response: We thank the Reviewer for making us aware of the missing information about the D29H-mutant of NRF2. We added a more detailed description of the mutant and the expected and observed results in our experiments. “*The D29H mutant diminishes interaction at one of the two critical binding sites between Nrf2 and Keap1 (DLG-motif) and hence protects Nrf2 from Keap1/Cul3-mediated degradation, resulting in a stabilization of Nrf2 protein levels⁶. Hence, treatment with SFN in this cell line is not expected to increase Nrf2 levels (as seen in Supplementary Figure 1B), but compound 7 is expected to block binding of Keap1 to Nrf2 that occurs via the still intact second binding site (ETGE-motif).*” (Page 4, lines 86-91).

Reviewer #3 (Remarks to the Author):

In this manuscript, the King laboratory optimizes a technique designed to detect the interaction of ectopically expressed (RT-Bind) or endogenously tagged (EndoBind) proteins of interest. The authors' methods basically follow the manufacture's (Promega Corporation) instruction (NanoBiT® Protein:Protein Interaction System) except for the use of pro-substrate, which used in other assay kit in this company, RealTime-Glo. This optimization is useful, however, the optimization does not use particularly novel technologies, but instead simply employs existing methods or materials. Moreover, although the advantage of using the pro-substrate is the detection of signal for longer time (pro-substrate: over 2 days), certainly fumirazine has a short half-life of about 2 hours, but other substrates provided for this interaction assay by Promega, Vivazine or Endurazine, have longer half-lives (about 10-20 hours or several days, respectively). Therefore, it does not appear to be novel unless there are dramatic advantages over these substrates.

Response: We appreciate the thoughtful comments from Reviewer #3 related to Vivazine and Endurazine, the Promega NanoBit substrates that have improved stability compared to fumirazine.

We evaluated both Vivazine and Endurazine in our RT-Bind assays. Consistent with Reviewer #3's assessment, Vivazine provided a good signal for several hours but was not useful for assays extending 24 hours or longer. Endurazine was similar to the pro-substrate RealTime-Glo in several aspects; good performance over a long period of time and the luminescent signals were comparable. We still prefer the pro-substrate RealTime-Glo for many of our assays because (1) the potential issue described by Walker et al (Walker et al., ACS Chem. Biol. 2017, 12, 1028–1037), regarding artifactual NanoLuc/NanoBit signal arising from non-specific protein aggregation due to lysed cells which likely is not relevant when using a pro-substrate, (2) RealTime-Glo is active for several days and this provides enhanced flexibility for assays developed for certain targets, and (3) pro-substrate RealTime-Glo is added to the cells at the time of seeding while Endurazine requires a media change and addition of the reagent on the first day of measurement, which can negatively impact the cells and results in more complicated screening workflows with lower data quality.

We realize that different readers will have a range of reactions regarding these results, which likely is linked to the particular NanoBit assays they are using in their laboratories. For many of them, fumirazine may be their best option, others may select Vivazine. But we feel strongly that the use of the pro-substrate RealTime-Glo as described in this manuscript describes additional flexibility to the NanoBit system that many readers will find important.

REVIEWERS' COMMENTS:

Reviewer #1 (Remarks to the Author):

All major issues have been properly addressed by the Authors, I have no further comments. I would only ask for some minor corrections regarding the editorial errors listed below:

1. Line 29: a period is missing at the end of the sentence.
2. Lines 52, 176 in the main text and line 15 in Supplementary Figure 2 caption: please unify the way of writing the name of the promoter.
3. Line 57: there should be "Keap/Cul3-mediated" instead of "Keap/Cul3 mediated".
4. Line 76: there should be "up to 24 hours" instead of "out to 24 hours".
5. Line 84: there should be "an Nrf2 mutant" instead of "a Nrf2-mutant".
6. Line 96: there should be "proteins of interest" instead of "proteins of interests".
7. Line 108: there should be "mutations" instead of "mutation".
8. Line 139: there should be "kinetics" instead of "kinetic".
9. Line 170: There should be "SmB" instead of "Sm".
10. Line 186: there should be "Life Technologies" instead of "LifeTechnologies". Besides, "2" in "CO₂" requires a subscript.
11. Line 194: there should be "LgB-tagged" instead of "Lg-tagged".
12. Line 196: there should be "µg/ml" instead of "ug/ml" after the numbers referring to the applied concentrations of antibiotics. Besides, there should be "SmB-tagged" instead of "Sm-tagged".
13. Lines 200 and 205: there should be "µg/ml" instead of "ug/ml".
14. Line 225: there should be "µg" instead of "ug".
15. Line 226: there should be "fractionation" instead of "fraction".
16. In Figure 2, the bar graphs are now partially masking the western blot images presented above.
17. Supplementary Figure 2 caption, line 17: there should be "PATU8988T" instead of "Patu8988T".
18. Supplementary Figure 2 caption, line 19: there should be "Note the upshift" instead of "Not the upshift".